# Numerical Block-Diagonalization and Linked-Cluster Expansion for Deriving Effective Hamiltonians: Applications to Spin Excitations

Tsutomu Momoi[1][⋆] and Owen Benton[2]

**1** RIKEN Center for Emergent Matter Science, Wako, Saitama, 351-0198, Japan
**2** School of Physical and Chemical Sciences, Queen Mary University of London, London, E1 4NS, United Kingdom

⋆ momoi@riken.jp

## Abstract

We present a numerical, non-perturbative framework for constructing effective Hamiltonians that accurately capture low-energy excitations in quantum many-body systems. The method combines block diagonalization based on the Cederbaum–Schirmer–Meyer transformation with the numerical linked-cluster expansion. A variational criterion is introduced to minimize changes in basis states within the low-energy subspace. This criterion uniquely determines the effective Hamiltonian and provides a robust guideline for selecting relevant eigenstates, even in the presence of avoided level crossings due to particle-number-nonconserving interactions. We validate the method using two spin models: the one-dimensional transverse-field Ising model and the two-dimensional Shastry–Sutherland model, relevant to $SrCu_2(BO_3)_2$. The resulting effective Hamiltonians successfully reproduce excitation dynamics, including the emergence of topological band structures. The framework is general, computationally feasible, and broadly applicable to a wide range of strongly correlated systems.

# 1   Introduction

In quantum many-body systems, low-energy physics is often governed by a restricted sub-space of the full Hilbert space. To understand the essential behavior in this regime, it is useful to construct effective Hamiltonians that accurately describe the dynamics within this subspace. A widely adopted strategy for deriving such models is to block-diagonalize the microscopic Hamiltonian via a unitary transformation, thereby decoupling the low-energy sector from higher-energy states.

Conventional approaches to block diagonalization are often based on perturbative methods, which provide analytic derivation of effective Hamiltonians through systematic series expansions [1–6]. While these techniques are conceptually transparent and computationally efficient in weak coupling regimes, their applicability is inherently limited by the requirement of a small expansion parameter. In many physically relevant situations, however, the system parameters lie outside the perturbative regime, making such approaches unreliable.

Several non-perturbative frameworks have been developed to overcome this limitation. A notable example is the continuous unitary (CU) transformation method [7–9], which employs differential flow equations to achieve block diagonalization without relying on perturbation theory. More recently, a hybrid method that combines the CU transformation with numerical linked-cluster (NLC) expansions [10–12] has been introduced to construct effective models on finite clusters and extrapolate them non-perturbatively to the thermodynamic limit [13, 14]. This approach has been successfully applied to both gapped systems and systems with quasi-degenerate low-energy manifolds [14–16]. These developments demonstrate the increasing demand for accurate and computationally tractable non-perturbative techniques.

It is important to recognize that block diagonalization is not unique. Different unitary transformations can yield distinct effective Hamiltonians, each suited to a different physical

description. For an overview of different approaches, see Ref. [17], which introduces a projective block diagonalization method combined with the NLC expansion. This intrinsic ambiguity poses challenges for both the physical interpretation and numerical realization of effective models. Therefore, selecting an appropriate transformation is crucial for constructing physically meaningful effective theories.

In this work, we present a numerically exact framework for deriving effective Hamiltonians based on a block diagonalization method initially proposed by Cederbaum, Schirmer, and Meyer (CSM) [18]. A key refinement to the original formulation is the introduction of a variational criterion that selects a unitary transformation which minimally transforms the low-energy subspace of interest. Under this condition, we show that the block-diagonalizing transformation becomes equivalent to the CSM transformation and is uniquely determined. The same criterion also provides a systematic guideline for selecting the relevant eigenstates, even in the presence of avoided level crossings or mixing caused by particle-number-nonconserving interactions, such as Dzyaloshinsky–Moriya (DM) couplings.

To extend the results to the thermodynamic limit, we combine the CSM-based block diagonalization with the NLC expansion, which systematically includes quantum corrections from larger clusters. This approach enables the construction of effective Hamiltonians using only low-energy eigenstates without requiring access to the full spectrum.

We demonstrate the effectiveness of this approach by applying it to two spin models. First, we apply it to the one-dimensional transverse-field Ising model, which is exactly solvable [19], and validate the results against exact magnon dispersion relations. Second, we investigate the two-dimensional Shastry–Sutherland model [20] with DM interaction relevant to the material $SrCu_2(BO_3)_2$ [21,22]. This compound exhibits gapful triplon excitations at low magnetic fields with weak DM-coupling-induced dispersion [23–32]. It also features a relatively large intradimer to interdimer exchange ratio, $J'/J = 0.6 \sim 0.63$ [33–35]. Although perturbative treatments have been widely applied in this parameter regime [34, 36–39, 39–41], their validity is not assured. We study triplon excitations and their topological band structures in the presence of DM interactions.

A notable strength of this framework is its robustness against spectral degeneracies and strong level mixing, which typically hinder perturbative approaches. By selecting an optimal low-energy basis variationally, our method yields unambiguous and physically meaningful effective Hamiltonians even in strongly correlated regimes. This capability opens a path toward the systematic and non-perturbative study of complex quantum systems.

The remainder of this paper is organized as follows. Section 2 presents the theoretical framework, including a refinement of the Cederbaum–Schirmer–Meyer transformation that enables unique and minimal block-diagonalization (Sec. 2.1). We also introduce a criterion for eigenstate selection that is robust in systems with level repulsion (Sec. 2.2). In Sec. 2.3, we describe the numerical linked-cluster (NLC) expansion and its application to constructing effective Hamiltonians, as well as their asymptotic behavior in gapped systems (Sec. 2.4). Sections 3 and 4 present applications of the method to two spin models: the one-dimensional transverse-field Ising model and the two-dimensional Shastry–Sutherland model with DM interactions. We demonstrate that the method accurately captures magnon and triplon excitations, including their topological band structures. Section 5 provides a summary and outlook, while technical details are presented in the Appendices.

## 2 Method

### 2.1 Block diagonalization and effective Hamiltonian

In this subsection, we present a method for block diagonalization based on the transformation introduced by Cederbaum, Schirmer, and Meyer (CSM) [18], and provide a practical extension that is suited for low-energy effective theories.

We consider a Hilbert space spanned by orthonormal basis states $|a_i\rangle$ for $i = 1, \cdots, N$ and a Hamiltonian $\mathcal{H} = \mathcal{H}_0 + \mathcal{H}_1$. The basis is partitioned into $m$ subspaces classified by the conserved quantum numbers of $\mathcal{H}_0$. The matrix representation $[\boldsymbol{H}]_{ij} = \langle a_i | \mathcal{H} | a_j \rangle$ is partitioned into $m \times m$ blocks, where $\mathcal{H}_0$ is block-diagonal and $\mathcal{H}_1$ introduces inter-block couplings.

Block-diagonalization of the Hermitian matrix $\boldsymbol{H}$ is performed using a unitary matrix $\boldsymbol{T}$:

$$
\boldsymbol{T}^\dagger \boldsymbol{H} \boldsymbol{T} = \begin{pmatrix} \boldsymbol{H}_{\text{eff},11} & & & 0 \\ & \boldsymbol{H}_{\text{eff},22} & & \\ & & \ddots & \\ 0 & & & \boldsymbol{H}_{\text{eff},mm} \end{pmatrix}, \tag{1}
$$

where each $\boldsymbol{H}_{\text{eff},ii}$ is an $n_i \times n_i$ Hermitian matrix. All off-diagonal blocks are null matrices. The transformed Hamiltonian corresponds to the Hamiltonian matrix in the new basis $|b_i\rangle = \sum_{j=1}^{N} [\boldsymbol{T}]_{ji} |a_j\rangle$ $(i = 1, \cdots, N)$, i.e., $\langle b_i | \mathcal{H} | b_j \rangle = [\boldsymbol{T}^\dagger \boldsymbol{H} \boldsymbol{T}]_{ij}$.

Since block diagonalization is not unique, CSM [18] proposed selecting the transformation that minimally changes the original basis, by imposing the minimization of the Euclidean norm

$$
\|\boldsymbol{T} - \boldsymbol{1}\| = \text{minimum}, \tag{2}
$$

where $\boldsymbol{1}$ denotes the identity matrix. They further proved that, under this condition, the optimal transformation is uniquely given by

$$
\boldsymbol{T} = \boldsymbol{S} \boldsymbol{S}_{\text{BD}}^\dagger (\boldsymbol{S}_{\text{BD}} \boldsymbol{S}_{\text{BD}}^\dagger)^{-1/2}, \tag{3}
$$

where $\boldsymbol{S}$ is a unitary matrix that diagonalizes the Hamiltonian,

$$
\boldsymbol{S}^\dagger \boldsymbol{H} \boldsymbol{S} = \boldsymbol{\Lambda}, \quad \boldsymbol{\Lambda} = \text{diag}(\lambda_1, \lambda_2, \ldots, \lambda_N), \tag{4}
$$

and the matrix $\boldsymbol{S}_{\text{BD}}$ is the diagonal-block part of $\boldsymbol{S}$ with respect to the subspace decomposition,

$$
\boldsymbol{S}_{\text{BD}} = \begin{pmatrix} \boldsymbol{S}_{11} & & & 0 \\ & \boldsymbol{S}_{22} & & \\ & & \ddots & \\ 0 & & & \boldsymbol{S}_{mm} \end{pmatrix}. \tag{5}
$$

In particular, each diagonal block $\boldsymbol{H}_{\text{eff},jj}$ of the transformed Hamiltonian can be constructed only with the eigenstates and eigenvalues of the corresponding subspaces. Using the singular value decomposition of $\boldsymbol{S}_{jj}$,

$$
\boldsymbol{S}_{jj} = \boldsymbol{U}_j \boldsymbol{\Sigma}_j \boldsymbol{V}_j^\dagger \tag{6}
$$

with $n_j \times n_j$ unitary matrices $\boldsymbol{U}_j$ and $\boldsymbol{V}_j$, and an $n_j \times n_j$ diagonal matrix $\boldsymbol{\Sigma}_j$ with positive entries, $\boldsymbol{H}_{\text{eff},jj}$ is given by

$$
\boldsymbol{H}_{\text{eff},jj} = \boldsymbol{U}_j \boldsymbol{V}_j^\dagger \boldsymbol{\Lambda}_j \boldsymbol{V}_j \boldsymbol{U}_j^\dagger, \tag{7}
$$

where $\boldsymbol{\Lambda}_j$ is the $j$th diagonal block of $\boldsymbol{\Lambda}$.

To construct a low-energy effective Hamiltonian, we extend the CSM framework by introducing a sector-specific minimization condition. Let the target low-energy subspace correspond to the first sector of the partitioned subspaces, with dimension $n$. The corresponding block $T_{11}$ of the unitary matrix $T$ acts within this subspace. We impose the following criterion:

**Criterion:** *The transformation implemented by the $n \times n$ matrix $T_{11}$ within the targeted subspace is chosen to minimized*

$$\|T_{11} - \mathbf{1}\| = \text{minimum}. \tag{8}$$

This condition ensures that the states within the targeted subspace remain as close as possible to their original forms and yields a natural effective Hamiltonian.

Based on this criterion, we now present a new theorem that uniquely determines the block-diagonalization method:

**Theorem:** *If the minimization condition in Eq. (8) is satisfied, then the matrix $T_{11}$ is uniquely given by*

$$T_{11} = U_1 \Sigma_1 U_1^\dagger \tag{9}$$

*and the effective Hamiltonian in the first subspace is*

$$H_{\text{eff},11} = U_1 V_1^\dagger \Lambda_1 V_1 U_1^\dagger. \tag{10}$$

The proof is given in Appendix A. The resulting effective Hamiltonian in Eq. (10) coincides with the CSM expression given in Eq. (7).

Although the proof of this theorem is a direct extension of the original CSM theorem, the result is particularly well suited for numerical applications. In Secs. 3 and 4, we demonstrate its use by applying the Lanczos algorithm to compute low-energy eigenstates. Once these eigenstates and eigenvalues are obtained, the matrix $T_{11}$ can be constructed using only this sector's data. This construction also enables the application of the target eigenstate selection criterion described in Sec. 2.2. Moreover, this minimal information is sufficient to determine the effective Hamiltonian in Eq. (10). Thus, the theoretical framework can be fully implemented using low-energy information alone, without requiring access to high-energy eigenstates.

## 2.2   Selection of target eigenstates

The criterion introduced in Eq. (8) not only determines the unitary transformation but also provides a practical guideline for selecting eigenstates in numerical calculations. In particular, it identifies the eigenstates that remain most closely aligned with the original basis states.

When the Hamiltonian contains only $\mathcal{H}_0$, the block-diagonal structure naturally arises from the sector decomposition of the Hilbert space. When $\mathcal{H}_1$ is introduced such that $\mathcal{H} = \mathcal{H}_0 + \lambda \mathcal{H}_1$ with a parameter $\lambda$, the eigenstates of interest become linear combinations of the original basis states $|a_j\rangle$ from both the first sector ($1 \leq j \leq n$) and the remaining sectors ($n + 1 \leq j \leq N$), with dominant support in the first sector.

As $\lambda$ increases, the original eigenstates $|a_j\rangle$ evolve into new basis states $|b_i\rangle = \sum_{j=1}^{N} [T]_{ji} |a_j\rangle$ for $i = 1, \cdots, n$. For sufficiently large $\lambda$, $\mathcal{H}_1$ induces strong hybridization between sectors, leading to avoided energy-level crossing and eventual interchange of eigenstates. In such cases, original states evolve into entirely different states, which is a well-known challenge in quasi-degenerate perturbation theory. The effective Hamiltonian must then be formulated based on the post-interchange eigenstates [14].

To perform eigenstate selection under these conditions, we adopt the minimization criterion in Eq. (8) as a selection guideline. For any candidate set of $n$ eigenstates, we construct the corresponding matrix $\boldsymbol{T}_{11}$ and evaluate the norm $\|\boldsymbol{T}_{11} - \mathbf{1}\|$. The set that minimizes this norm is selected, as it corresponds to eigenstates most faithfully aligned with the original low-energy basis.

Section 4 demonstrates this selection process in a system exhibiting avoided level crossings and state reordering. Figure 4 illustrates these processes.

## 2.3 Numerical linked-cluster expansion

For a finite-size cluster $c$, the effective Hamiltonian can be obtained by performing block diagonalization with a suitably chosen set of eigenstates. The derived Hamiltonian includes interaction terms whose magnitudes are cluster-specific physical quantities. To extrapolate these quantities to the thermodynamic limit, we employ the numerical linked-cluster (NLC) expansion [10–12], following the procedure outlined in Ref. [13]. For completeness, we briefly summarize the method below.

We consider a model defined on a lattice with $N$ subunits (e.g., sites, dimers, or tetrahedra), and seek to compute the expectation value of an extensive observable $\mathcal{O}$ per subunit, $\frac{1}{N}\langle\mathcal{O}\rangle_c$, where $\langle\mathcal{O}\rangle_c$ denotes the value obtained from a finite cluster $c$ with open boundary conditions. In the NLC approach, the $n$th-order estimate of this quantity is given by

$$\frac{1}{N}\langle\mathcal{O}\rangle_{\mathrm{NLC}n} = \sum_{c,\mathrm{size}(c)\leq n} l_c W_c, \tag{11}$$

where the summation runs over all inequivalent connected clusters up to size $n$. Here, the cluster multiplicity $l_c$ denotes the number of embeddings of cluster $c$ per subunit in the infinite lattice, and $W_c$ is the cluster weight, defined recursively as

$$W_c = \langle\mathcal{O}\rangle_c - \sum_{s\in c} W_s, \tag{12}$$

where the summation runs over all subclusters $s$ of $c$.

Here, the quantities we are interested in are matrix elements of the effective Hamiltonian. For example to find the NLC estimate of the nearest-neighbor hopping amplitude we would define an extensive quantity on each cluster $c$

$$\mathcal{T}_{1,c} = \sum_i \sum_{j\in\mathrm{nn}_i} [\boldsymbol{H}_{\mathrm{eff},c}]_{ij}.$$

where the inner sum is over nearest-neighbors of subunit $i$.

The NLC estimate of $\frac{\mathcal{T}_1}{N}$ is then given by Eqs. (11-12) with $\mathcal{T}_1$ in the place of $\langle\mathcal{O}\rangle$ and the hopping amplitude is:

$$t_1 = \frac{\mathcal{T}_1}{zN}$$

where $z$ is the coordination number.

## 2.4 Asymptotic forms of effective Hamiltonians in gapped systems

Before applying the numerical framework to specific models, we analyze the asymptotic behavior of effective Hamiltonians for excited states in systems with a unique ground state and a finite excitation gap. We focus on the single-excitation sector in a $d$-dimensional hypercubic lattice. The effective Hamiltonian is given by

$$\mathcal{H}_{\mathrm{eff}} = \sum_{i,j} t_{ij} a_i^\dagger a_j, \tag{13}$$

where $t_{ij} = t_{ji}^*$, and $a_i^\dagger$ ($a_i$) denotes the creation (annihilation) operator of an excitation at site $i$. The ground state is defined as the vacuum. The dispersion relation $\varepsilon(\boldsymbol{k})$ is related to $t_{ij}$ by

$$\varepsilon(\boldsymbol{k}) = \sum_{r_i - r_j} t_{ij} \exp\{i\boldsymbol{k} \cdot (\boldsymbol{r}_i - \boldsymbol{r}_j)\}. \tag{14}$$

To characterize the spatial decay of $t_{ij}$ near and away from criticality, we consider a dispersion of the form

$$\varepsilon(\boldsymbol{k}) = \left[ 2\left( d - \sum_{i=1}^{d} \cos k_i \right) + m^{2/Z} \right]^{Z/2}, \tag{15}$$

where $Z$ is the dynamical exponent and $m$ controls the energy gap, satisfying $\varepsilon(\boldsymbol{0}) = m$. At the critical point ($m = 0$), the low-energy dispersion behaves as $\varepsilon(\boldsymbol{k}) \simeq |\boldsymbol{k}|^Z$.

The Fourier transform of $\varepsilon(\boldsymbol{k})$ gives the asymptotic behavior of $t_{ij}$:

$$t_{ij} \propto \begin{cases} \exp(-m|\boldsymbol{r}_j - \boldsymbol{r}_i|), & (|\boldsymbol{r}_j - \boldsymbol{r}_i| \gg m^{-1}), \\ |\boldsymbol{r}_j - \boldsymbol{r}_i|^{-d-Z}, & (|\boldsymbol{r}_j - \boldsymbol{r}_i| \ll m^{-1}), \end{cases} \tag{16}$$

for $Z \neq 2$, and

$$t_{ij} \propto \sum_{\mu=1,2,3} (\delta_{r_i - r_j, e_\mu} + \delta_{r_i - r_j, -e_\mu}), \tag{17}$$

for $Z = 2$, where $\boldsymbol{e}_\mu$ are the unit vectors of the lattice.

These results demonstrate that, in gapped systems, the effective couplings decay rapidly with distance. Consequently, the NLC expansions converges quickly even for small clusters, enabling accurate and efficient construction of effective Hamiltonians.

# 3   Application I: One-dimensional transverse-field Ising model

As a first application, we examine magnon excitations in the one-dimensional transverse-field Ising model. The Hamiltonian is defined as

$$\mathcal{H}_{1d} = -\sum_{i=1}^{n} \sigma_i^z - J \sum_{i=1}^{n-1} \sigma_i^x \sigma_{i+1}^x, \tag{18}$$

where $\sigma_i^\alpha$ denotes the Pauli matrix at site $i$, and the second term describes nearest-neighbor interactions in an $n$-spin chain $c$ with open boundary conditions.

We consider the parameter range $0 \leq J \leq 1$. For $0 \leq J < 1$, the ground state remains unique and polarized along the field direction. This exactly solvable model [19] has previously served as a benchmark for the continuous unitary transformation approach [13]. Our results are compared directly with earlier findings, as discussed in Appendix B.

We partition the Hilbert space into three sectors: (i) the fully polarized state, (ii) an $n$-dimensional space of single spin-flip states, and (iii) the remaining higher-excitation space. The Hamiltonian is block-diagonalized accordingly.

In the first sector, the diagonal element of the block-diagonalized Hamiltonian matrix reduces to the ground-state energy:

$$\mathcal{H}_{\text{eff}}^{(0)} = E_0(n). \tag{19}$$

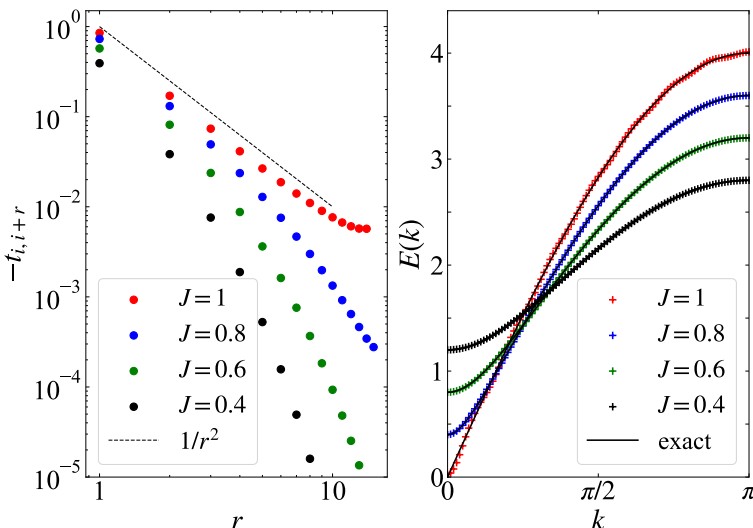

Figure 1: Numerical results obtained from block diagonalization and NLC expansion. (Left) Distance dependence ($r$-dependence) of the magnon hopping amplitudes $t_{i,i+r}^{\mathrm{NLC}}$. (Right) One-magnon excitation spectra compared with exact results.

In the second single-spin-flip sector, the effective Hamiltonian takes the form

$$\mathcal{H}_{\mathrm{eff}}^{(1)} = E_0(n) + \sum_{i,j \in c} t_{ij}(n) a_i^\dagger a_j, \tag{20}$$

where $t_{ij}(n) = t_{ji}(n) \in \mathbb{R}$, and $a_i$ ($a_i^\dagger$) is an annihilation (creation) operator of a single-spin flip at site $i$.

We evaluate $E_0(n)$ and the hopping amplitudes $\sum_{i=1,\cdots,n-r} t_{i,i+r}(n)$ for $r = 0, \ldots, n-1$ for each cluster, and applied the NLC expansion to the quantities $E_0(n)/n$ and $t_{i,i+r}(n)$ for cluster size up to $n = 16$.

Figure 1 shows the estimated hopping amplitudes $t_{i,i+r}^{\mathrm{NLC}}$, as a function of $r$, and the corresponding one-magnon excitation spectrum. For $J < 1$, the hopping amplitudes $t_{i,i+r}^{\mathrm{NLC}}$ decay exponentially with $r$, consistent with a finite energy gap. As $J$ approaches 1, the decay becomes slower, and, at $J = 1$, the decay follows a $1/r^2$ power law, corresponding to a gapless excitation spectrum with dynamical exponent $Z = 1$.

Despite the small system size (up to 16 spins), the resulting excitation spectra obtained agree well with exact results. This is consistent with previous studies [13].

## 4  Application II: Two-dimensional Shastry–Sutherland model with Dzyaloshinsky–Moriya couplings

As a second application, we examine excitation behavior in the two-dimensional Shastry–Sutherland model [20], incorporating Dzyaloshinsky–Moriya (DM) interactions. This model captures the essential features of the spin-dimer compound $SrCu_2(BO_3)_2$ [21].

### 4.1  Models

#### 4.1.1  Microscopic spin model

The model is defined on a two-dimensional lattice consisting of orthogonal $A$- and $B$-type dimers (Fig. 2). The Hamiltonian is given by $\mathcal{H}_{2d} = \mathcal{H}_0 + \mathcal{H}_1$, where $\mathcal{H}_0$ describes intradimer

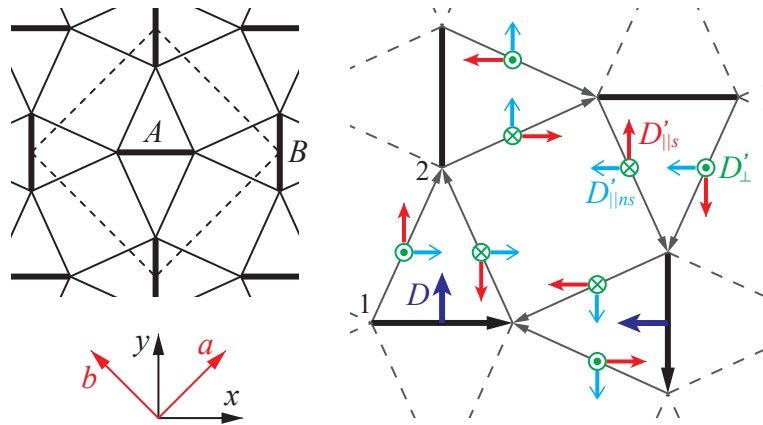

Figure 2: (Left) Unit cell of the Shastry–Sutherland lattice. (Right) Symmetry-constrained DM vectors in $SrCu_2(BO_3)_2$, consistent with $S_4$ and $C_{2V}$ crystal symmetries. The site indices of $\boldsymbol{D}_{ij}$ and $\boldsymbol{D}'_{ij}$ are assigned following the direction $i \rightarrow j$ indicated by the arrows on the bonds.

interactions:

$$\mathcal{H}_0 = \sum_{\langle i,j \rangle_1} (J\boldsymbol{S}_i \cdot \boldsymbol{S}_j + \boldsymbol{D}_{ij} \cdot \boldsymbol{S}_i \times \boldsymbol{S}_j) - h \sum_j S_j^z, \tag{21}$$

and $\mathcal{H}_1$ represents interdimer couplings:

$$\mathcal{H}_1 = \sum_{\langle i,j \rangle_2} (J'\boldsymbol{S}_i \cdot \boldsymbol{S}_j + \boldsymbol{D}'_{ij} \cdot \boldsymbol{S}_i \times \boldsymbol{S}_j). \tag{22}$$

Here, $\langle i,j \rangle_n$ denotes $n$th-neighbor spin pairs.

The DM vectors $\boldsymbol{D}_{ij}$ and $\boldsymbol{D}'_{ij}$ respect the $I\bar{4}2m$ space group symmetries [42] of $SrCu_2(BO_3)_2$, including the $S_4$ symmetry centered on the square plaquettes of four dimers, and the $C_{2V}$ symmetry centered on individual dimers. The symmetry allowed DM vector configurations are shown in Fig. 2. We adopt the parameter set $\boldsymbol{D}_{ij}/J = (0, 0.048, 0)$ for $A$-type dimers and $\boldsymbol{D}'_{ij}/J \equiv (D'_{\|ns}, D'_{\|s}, D'_{\perp}) = (0.005, 0.008, 0.014)$ for the $(1,2)$-bond shown in Fig. 2, following *ab initio* calculations [35].

### 4.1.2 Effective triplon model

In the low-field spin-gap phase, excitations are described by spin-1 triplons. Although DM interactions break the triplon number conservation, we partition the Hilbert space by triplon number and derive the effective Hamiltonian in the one-triplon sector.

Each dimer is characterized by its center position $\boldsymbol{r}$, with its two spins labeled as sites 1 and 2. In the absence of DM interactions, the ground state is a gapped singlet state [20]. Triplon excitations are created or annihilated by triplon operators: $t_{\boldsymbol{r}}^{\mu\dagger} = iS_{\boldsymbol{r},1}^\mu - iS_{\boldsymbol{r},2}^\mu$, $t_{\boldsymbol{r}}^\mu = -iS_{\boldsymbol{r},1}^\mu + iS_{\boldsymbol{r},2}^\mu$, for $\mu = x, y, z$, where $\boldsymbol{r} \in \Lambda$, the lattice of all dimer centers.

The effective one-triplon Hamiltonian has the form

$$\mathcal{H}_{\text{eff}} = \sum_{\alpha=A,B} \sum_{\boldsymbol{r} \in \Lambda_\alpha} \left( \sum_{\boldsymbol{\delta}} t_{\boldsymbol{r}+\boldsymbol{\delta}}^\dagger \boldsymbol{M}_{\alpha\alpha,\boldsymbol{\delta}} t_{\boldsymbol{r}} + \sum_{\boldsymbol{\delta}'} t_{\boldsymbol{r}+\boldsymbol{\delta}'}^\dagger \boldsymbol{M}_{\bar{\alpha}\alpha,\boldsymbol{\delta}'} t_{\boldsymbol{r}} \right), \tag{23}$$

where $\boldsymbol{t}_{\boldsymbol{r}} = (t_{\boldsymbol{r}}^x, t_{\boldsymbol{r}}^y, t_{\boldsymbol{r}}^z)^T$, and $\boldsymbol{M}_{\alpha\beta,\boldsymbol{\delta}}$ are $3 \times 3$ hopping matrices from $\beta$-type to $\alpha$-type dimers. We define $\bar{A} = B$ and $\bar{B} = A$. The sublattices $\Lambda_A$ and $\Lambda_B$ refer to the sets of $A$- and $B$-type dimer

260 centers, respectively. The vectors $\boldsymbol{\delta}$ ($\boldsymbol{\delta}'$) denote relative positions within (between) sublattices.
261 Hermiticity imposes the condition $\boldsymbol{M}^{\dagger}_{\alpha\beta,\boldsymbol{\delta}} = \boldsymbol{M}_{\beta\alpha,-\boldsymbol{\delta}}$.

262      Lattice symmetries impose further constraints on the matrices $\boldsymbol{M}_{\alpha\beta,\boldsymbol{\delta}}$ (see Appendix C). For
263 instance, the on-site potential $\boldsymbol{M}_{AA,(0,0)}$ takes the form

$$M_{AA,(0,0)} = \begin{bmatrix} R_{01} & I_{01} & 0 \\ -I_{01} & R_{02} & 0 \\ 0 & 0 & R_{03} \end{bmatrix}, \tag{24}$$

264 where $R_i$ and $I_i$ are real and imaginary coefficients, respectively, and the nearest-neighbor
265 hopping matrix $\boldsymbol{M}_{BA,(1,0)}$ is given by

$$M_{BA,(1,0)} = \begin{bmatrix} I_{11} & R_{11} & I_{12} \\ R_{12} & I_{13} & R_{13} \\ I_{14} & R_{14} & I_{15} \end{bmatrix}. \tag{25}$$

266 We include triplon hopping terms up to second and third neighbors.

## 4.2   Numerical results

### 4.2.1   NLC expansion

269 We computed the matrix elements of the on-site potential and hopping terms using the NLC
270 expansion. For each interaction type, the expansion begins with the smallest connected clus-
271 ter containing the all relevant sites and systematically incorporates contributions from larger
272 clusters up to a chosen truncation order. For on-site terms, the expansion starts from a single
273 dimer. Higher-order corrections are incorporated through contributions from larger clusters
274 using the cluster weights described in Eq. (11). For nearest-neighbor dimer interactions, the
275 minimal cluster consists of two dimers connected by two interdimer bonds, with additional
276 contributions from larger clusters. For second-neighbor interactions, the expansion starts from
277 three dimers connected through interdimer couplings. Larger clusters were constructed by first
278 defining a basic unit block consisting of four dimers arranged in a square and by combining
279 them through edge-sharing or corner-sharing. To systematically generate subclusters used in
280 the NLC expansion [Eqs. (11) and (12)], we applied a bond-dilution algorithm, successively
281 removing selected pairs of interdimer bonds. Calculations were performed on clusters con-
282 taining up to two four-dimer blocks, consisting of seven dimers and 14 spins.

283      To evaluate deviations from perturbative results, we introduce a scaling factor $\lambda$ for the
284 interdimer term and analyze the $\lambda$-dependence of the matrix elements. The deformed Hamil-
285 tonian is given by

$$\mathcal{H}_{2d}(\lambda) = \mathcal{H}_0 + \lambda\mathcal{H}_1. \tag{26}$$

286 The original Hamiltonian for $SrCu_2(BO_3)_2$ corresponds to $\lambda = 1$. Figure 3 presents the $\lambda$
287 and cluster-size dependence of selected matrix elements of the on-site potential and nearest-
288 neighbor hopping. Numerical data for second- and third-neighbor hoppings are presented in
289 Appendix D.

290      At $\lambda = 1$, the real parts of the on-site potential and nearest-neighbor hopping terms show
291 significant renormalization, while their imaginary parts remain nearly unchanged. Matrix
292 elements associated with second- and third-neighbor hopping are generally negligible, except
293 for weak enhancements in the diagonal elements of second-neighbor hopping matrices near
294 $\lambda = 1$.

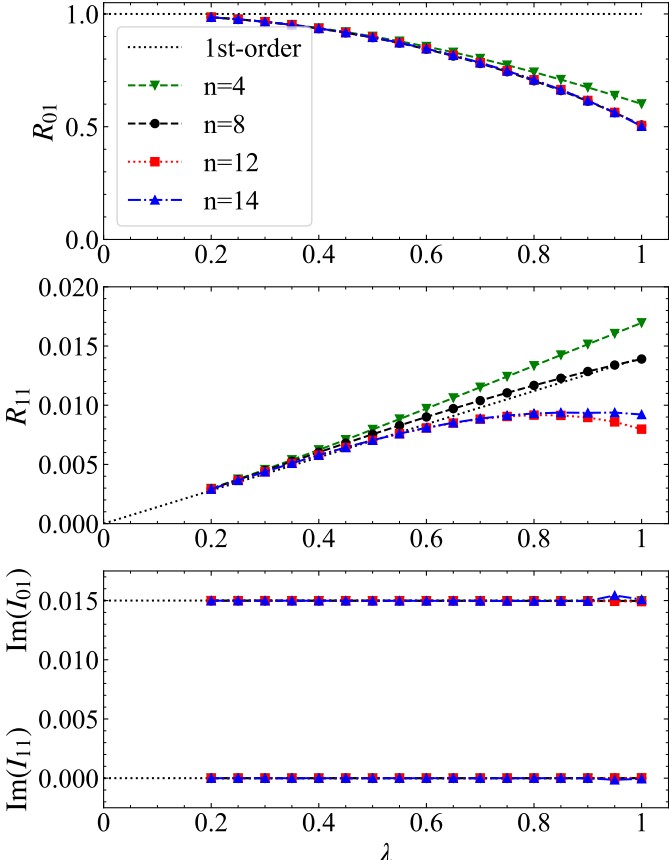

Figure 3: $\lambda$-dependence of selected matrix elements of on-site potential ($R_{01}$ and $I_{01}$) and nearest-neighbor hopping ($R_{11}$ and $I_{11}$) for $J'/J = 0.6$ and $h/J = 0.015$. Dashed lines represent the first-order perturbative expansion results [30]. Here, $n$ denotes the number of spins in the largest clusters used for the NLC expansion.

### 4.2.2 Eigenstate selection

Due to DM interactions, the triplon number is not conserved, leading to avoided level crossings and eigenstate interchange. To consistently construct the effective Hamiltonian, eigenstates must be selected according to the criterion defined in Eq. (8), as described in Sec. 2.2.

Figure 4 shows a typical example of eigenstate selection, where level repulsion and interchange occur between one-triplon and two-triplon states. At each repulsion point, two eigenstates exchange character. The selection criterion identifies the eigenstates that retain the character of the original low-energy basis. Because the selected eigenstates may change discontinuously, this selection process can introduce cusp-like anomalies in physical quantities. Although a method for suppressing such anomalies exists within the CU transformation framework [14], no comparable method is currently available for other unitary transformation approaches [17].

These anomalies typically involve the exchange of a single pair of eigenstates, and their amplitude does not scale with system size $n$. In practice, the NLC expansion significantly reduces these anomalies, yielding smoother physical quantities. In our numerical implementation, such anomalies were successfully suppressed by incorporating many subclusters with small size differences, systematically generated using the bond-dilution method.

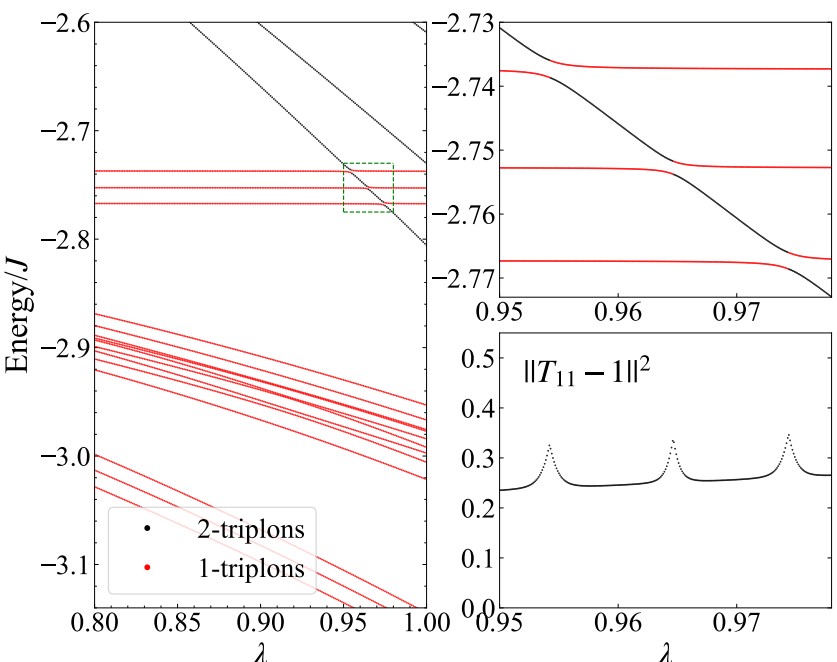

Figure 4: (Left) $\lambda$-dependence of the energy spectrum for a 10-spin cluster. Red dots correspond to single-triplon excitations, and black dots represent two-triplons states. (Right Top) Enlarged view of the green dashed region from the left panel. Following level repulsion, the states interchange, with red dots indicating the eigenstates selected according to the criterion in Eq. (8). (Right Bottom) Plot of $\|T_{11} - 1\|^2$ for the selected eigenstates.

### 4.2.3 Triplon bands

Figure 5 presents the excitation-energy spectra of triplon bands for $J'/J = 0.3$ and $J'/J = 0.62$ under different magnetic fields. The unit cell contains two dimers, resulting in six triplon bands. Due to crystal symmetries, Kramers degeneracy is enforced along the Brillouin zone boundary, where two bands remain connected. This behavior is visible in the X-M segment of the figure. This degeneracy is proven in Appendix E. Consequently, the excitation spectrum consists of three connected bands.

For $J' = 0.3J$, the triplon spectrum exhibits a nearly flat central band with the upper and lower bands symmetrically placed around it, in agreement with perturbative expansion results [30, 31]. In contrast, for $J' = 0.62J$, the central band develops a significant dispersion, and the band symmetry is lost. This is consistent with inelastic neutron scattering observations [32]. While previous studies [30, 32] introduced phenomenological second-neighbor hopping to account for experimental results, our non-perturbative method reproduces these features naturally using only nearest-neighbor couplings.

We also computed the Berry curvature $\Omega_n(\boldsymbol{k})$ numerically [43] using $\Omega_n(\boldsymbol{k}) = \partial_x A_n^y(\boldsymbol{k}) - \partial_y A_n^x(\boldsymbol{k})$, where $A_n^\mu(\boldsymbol{k}) = i\langle n(\boldsymbol{k})|\partial_\mu|n(\boldsymbol{k})\rangle$. Here, $\partial_\mu = \frac{\partial}{\partial k_\mu}$, and $|n(\boldsymbol{k})\rangle$ is the normalized eigenstate of the $n$th Bloch band. The Chern number of the $n$th band was evaluated from

$$c_n = -\frac{1}{2\pi} \int d^2k\, \Omega_n(\boldsymbol{k}). \tag{27}$$

Topological triplon bands with non-zero Chern numbers appear for small magnetic fields ($h/J \lesssim 0.02$). At $J'/J = 0.3$, we find $(c_1, c_2, c_3) = (2, 0, -2)$ in this range and, at stronger fields, trivial bands $(0, 0, 0)$ appear, consistent with perturbative results [30, 32]. At $J'/J = 0.62$, the

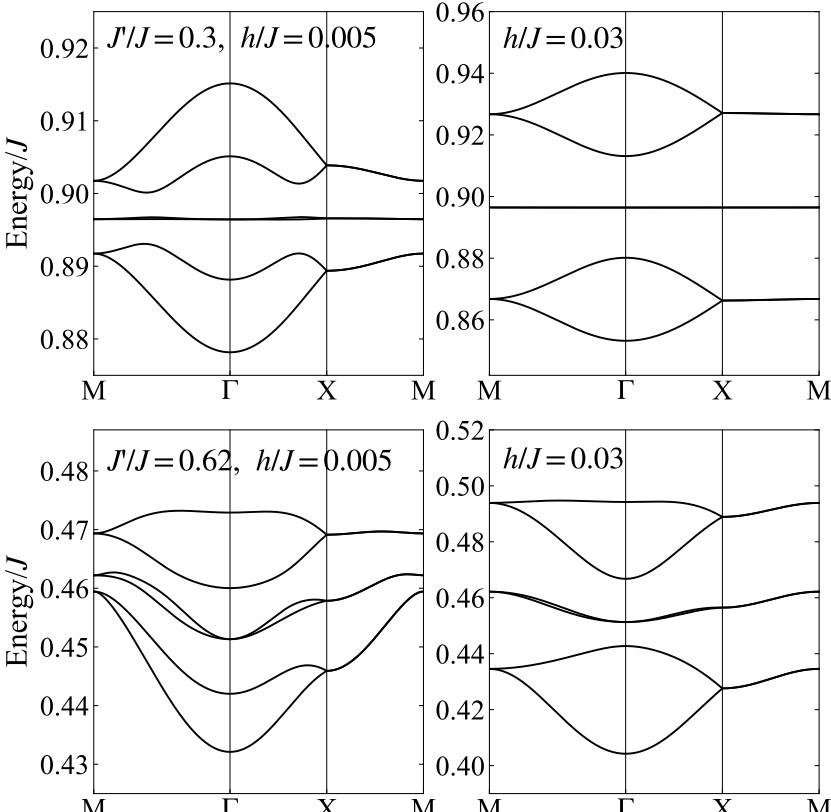

Figure 5: Excitation-energy spectra derived from the triplon effective Hamiltonians for $J' = 0.3J$ (upper panels) and $J' = 0.62J$ (lower panels). At $h = 0.005J$, the triplon bands are topological, characterized by Chern numbers $(c_1, c_2, c_3) = (2, 0, -2)$, ordered from the lowest to highest band in both cases. At $h = 0.03J$, all bands are topologically trivial. High-symmetry points in the Brillouin zone are defined as $\Gamma = (k_x, k_y) = (0, 0)$, $X = (\pi/2, \pi/2)$, and $M = (\pi, 0)$, where the momentum space is defined in units of the inverse nearest-neighbor dimer distance.

region with $(2, 0, -2)$ becomes narrower and intermediate topological bands such as $(2, -1, -1)$, $(0, 1, -1)$, and $(1, 1, -2)$ emerge. These intermediate regions are absent in perturbative treatments. The thermal Hall conductivity computed at $h/J = 0.01$ from the effective Hamiltonian, assuming a bare coupling $J = 7$ meV, shows reasonable agreement with first-order perturbative estimates based on an effective coupling $J = 3.8$ meV, for temperatures below 10 K [44].

# 5    Summary and methodological outlook

We developed a non-perturbative framework for constructing effective Hamiltonians in quantum many-body systems. Our approach combines numerical block diagonalization with the numerical linked-cluster (NLC) expansion, enabling the direct construction of effective low-energy models from microscopic Hamiltonians. It is based on a refinement of the Cederbaum–Schirmer–Meyer (CSM) transformation, incorporating a variational criterion that selects a unitary transformation minimizing its action within the target low-energy subspace. This criterion uniquely determines the transformation and provides a systematic guideline for identifying physically relevant eigenstates, even in the presence of strong level repulsion or mixing. As a result, the resulting effective Hamiltonian is both unambiguous and physically meaningful.

We demonstrated the utility of this framework by applying it to two spin models. In the one-dimensional transverse-field Ising model, it accurately reproduced the magnon dispersion, matching exact results even for small system sizes. In the two-dimensional Shastry–Sutherland model with Dzyaloshinsky–Moriya interactions, it captured the non-perturbative renormalization of hopping amplitudes and the emergence of topological triplon band structures.

A major advantage of our method is that it requires only low-energy eigenstates and does not depend on high-energy data. The variational criterion ensures robustness even in parameter regimes where perturbation theory fails due to avoided level crossings and eigenstate interchange. The NLC expansion captures quantum correlations effectively from large clusters, enabling the effective Hamiltonians to reflect long-range couplings and topological features without requiring phenomenological input. Taken together, the method is computationally tractable and broadly applicable.

This framework can be extended in several directions. At finite temperatures, it can be used to calculate thermodynamic observables and transport properties such as the thermal Hall conductivity. By incorporating transformed operators, it may also be applied to the evaluation of dynamical response functions, including spectral densities and neutron-scattering cross sections.

The method is also promising for analyzing fractionalized excitations, such as those found in Kitaev magnets and quantum spin liquids. In such systems, non-perturbative effective Hamiltonians could yield new insights into spin fractionalization and topological quasiparticles. Although the present study has focused on gapped systems, prior works using the CU transformation [15, 16, 45] have shown that the NLC expansion is effective even in systems with ground-state degeneracy. Thus, our framework is expected to be applicable to gapless or symmetry-broken systems, further broadening its applicability to a wide class of correlated quantum systems.

# Acknowledgements

The authors thank Hiroshi Ueda for bringing Ref. [18] to their attention. The authors are grateful to Kai Phillip Schmidt for providing numerical data from Ref. [13] and useful comments. They also acknowledge stimulating discussions with Hiroshi Ueda, Shingo Kobayashi,

376   and Nic Shannon.

377   **Funding information**   This work was supported by KAKENHI Grant No. JP20K03778 from
378   the Japan Society for the Promotion of Science (JSPS).

## A   Proof of the Theorem in Sec. 2.1

380   This appendix provides a proof of the theorem presented in Sec. 2.1.

381      We begin by introducing the unitary matrix $F$ that diagonalizes the block-diagonal matrix
382   $T^\dagger H T$ such that $F T^\dagger H T F^\dagger = \Lambda$. The matrix $F$ is block diagonal, with each diagonal block
383   $F_{jj}$ $(j = 1, \cdots, m)$ being unitary. Consequently, $T$ can be expressed as $T = SF$, leading to
384   $T_{11} = S_{11} F_{11}$ for the $(1, 1)$ block.

385      The squared Euclidean norm of $T_{11} - \mathbf{1}$ is then written as

$$
\begin{aligned}
\|T_{11} - \mathbf{1}\|^2 &= \mathrm{Tr}(T_{11}^\dagger - \mathbf{1})(T_{11} - \mathbf{1}) \\
&= \mathrm{Tr}\,\mathbf{1} - \mathrm{Tr}(T_{11} + T_{11}^\dagger) + \mathrm{Tr}\,T_{11}^\dagger T_{11} \\
&= n - \mathrm{Tr}(S_{11} F_{11} + S_{11}^\dagger F_{11}^\dagger) + \mathrm{Tr}\,S_{11}^\dagger S_{11}.
\end{aligned}
$$

386   The key difference from the proof of the CSM theorem, which considers the full unitary matrix
387   $T$, lies in the final term $\mathrm{Tr}\,T_{11}^\dagger T_{11}$.

388      We now apply singular value decomposition (SVD) to the block matrix $S_{11}$:

$$
S_{11} = U_1 \Sigma_1 V_1^\dagger, \tag{A.1}
$$

389   where $U_1$ and $V_1$ are $n \times n$ unitary matrices, and $\Sigma_1$ is an $n \times n$ diagonal matrix. The phases
390   of the column vectors of $U_1$ and $V_1$ are chosen so that the diagonal entries $\sigma_j$ of $\Sigma_1$ are real
391   and non-negative, $\sigma_j \geq 0$ for $j = 1, \cdots, n$. Furthermore, we assume the singular values are
392   ordered non-increasingly, $\sigma_1 \geq \sigma_2 \geq \cdots \geq \sigma_n \geq 0$. Using this decomposition, we find

$$
\mathrm{Tr}\,S_{11} F_{11} = \mathrm{Tr}\,\Sigma_1 V_1^\dagger F_{11} U_1 = \sum_{j=1}^{n} \sigma_j [V_1^\dagger F_{11} U_1]_{jj},
$$

$$
\mathrm{Tr}\,S_{11}^\dagger S_{11} = \mathrm{Tr}\,\Sigma_1^\dagger \Sigma_1 = \sum_{j=1}^{n} \sigma_j^2.
$$

393   Therefore, the squared norm becomes

$$
\|T_{11} - \mathbf{1}\|^2 = n - 2\sum_{j=1}^{n} \sigma_j \mathrm{Re}[V_1^\dagger F_{11} U_1]_{jj} + \sum_{j=1}^{n} \sigma_j^2. \tag{A.2}
$$

394   Since the SVD is unique [46], the squared norm is minimized when the real part of each
395   diagonal element $[V_1^\dagger F_{11} U_1]_{jj}$ is maximized. Because $V_1^\dagger F_{11} U_1$ is unitary, the absolute value
396   of every matrix element is at most 1. Hence, the norm $\|T_{11} - \mathbf{1}\|$ achieves its minimum only
397   when $V_1^\dagger F_{11} U_1$ is the $n \times n$ identity matrix. This implies that the condition in Eq. (8) is satisfied
398   when $F_{11} = V_1 U_1^\dagger$ and $T_{11} = S_{11} F_{11} = U_1 \Sigma_1 U_1^\dagger$, which proves Eq. (9).

399      Furthermore, since $F_{11} H_{\mathrm{eff},11} F_{11}^\dagger = \Lambda_1$, the effective Hamiltonian is expressed as

$$
H_{\mathrm{eff},11} = F_{11}^\dagger \Lambda_1 F_{11} = U_1 V_1^\dagger \Lambda_1 V_1 U_1^\dagger, \tag{A.3}
$$

400   confirming Eq. (10) and completing the proof.

## B  Comparison of CSM and CU transformations in block diagonalization

Two effective Hamiltonians derived using different unitary transformations are related through an additional unitary transformation. This appendix compares two approaches: the CSM transformation and the continuous unitary (CU) transformation. We analyze the effective Hamiltonians for the one-dimensional transverse-field Ising model discussed in Sec. 3 and compare our results with those obtained using the CU transformation in Ref. [13].

The sum of the on-site potential, $\sum_i t_{ii}(n)$, corresponds to the trace of the effective Hamiltonian, $\sum_{i=1}^{n} t_{ii}(n) = \text{Tr} H_{\text{eff},11}$. This quantity is invariant under unitary transformations. As a result, the on-site potential $t_0^{\text{NLC}}(n)$, derived from the NLC expansion of $t_{ii}(n)$ up to $n$ spins, is independent of the specific block-diagonalization method employed. This invariance is confirmed by the data shown in Fig. 6(Upper).

In contrast, the hopping amplitudes $t_r^{\text{NLC}}(n)$, extracted from the NLC expansion of the off-diagonal components $[H_{\text{eff},11}]_{i,i+r}$, do depend on the choice of transformation. Figure 6(Lower) shows that the CSM transformation yields faster convergence of the energy gap,

$$\Delta^{\text{NLC}}(n) = t_0^{\text{NLC}}(n) + \sum_{r=1}^{n} t_r^{\text{NLC}}(n), \tag{B.1}$$

to the exact value compared to the CU transformation.

## C  Symmetry constraints on the triplon hopping matrices for SrCu$_2$(BO$_3$)$_2$

SrCu$_2$(BO$_3$)$_2$ exhibits $S_4$ symmetry around the center of each square formed by four dimers, as well as $C_{2V}$ symmetry around the center of each dimer. The $S_4$ operation consists of a $\pi/2$ rotation around the $z$-axis combined with a mirror reflection in the $xy$-plane. The $C_{2V}$ symmetry includes a $\pi$-rotation around the $z$-axis $C_2(z)$, a mirror reflection in the $xz$-plane $\sigma_{xz}$, and a mirror reflection in the $yz$-plane $\sigma_{yz}$. In a magnetic field applied along the $z$ axis, the system retains symmetry under the following operations: (1) $S_4$ around the center of each square composed of four adjacent dimers; (2) $C_2(z)$, $U_T \Theta \sigma_{xz}$, and $U_T \Theta \sigma_{yz}$ about the center of each dimer. Here, $U_T = \otimes_{r \in \Lambda, m=1,2}(-i\sigma_{r,m}^x)$ and $\Theta$ denotes complex conjugation.

The operations $U_T \Theta \sigma_{yz}$ and $C_2(z)$ transform the triplon operators as $t_r \rightarrow R_{\sigma(x)} t_r$ and $t_r \rightarrow R_{\pi(z)} t_r$, respectively, for $r \in \Lambda_A$, with

$$R_{\sigma(x)} = \begin{bmatrix} -1 & 0 & 0 \\ 0 & 1 & 0 \\ 0 & 0 & 1 \end{bmatrix}, \quad R_{\pi(z)} = \begin{bmatrix} 1 & 0 & 0 \\ 0 & 1 & 0 \\ 0 & 0 & -1 \end{bmatrix}. \tag{C.1}$$

The $S_4$ operation transforms $t_r \rightarrow S_{\alpha \rightarrow \tilde{\alpha}} t_{r'}$ for $r \in \Lambda_\alpha$ and $r' \in \Lambda_{\tilde{\alpha}}$, with

$$S_{A \rightarrow B} = \begin{bmatrix} 0 & 1 & 0 \\ -1 & 0 & 0 \\ 0 & 0 & -1 \end{bmatrix}, \quad S_{B \rightarrow A} = \begin{bmatrix} 0 & -1 & 0 \\ 1 & 0 & 0 \\ 0 & 0 & 1 \end{bmatrix}. \tag{C.2}$$

The on-site potential $t_r^\dagger M_{AA,(0,0)} t_r$ ($r \in \Lambda_A$) is invariant under $C_2(z)$, $U_T \Theta \sigma_{xz}$, and $U_T \Theta \sigma_{yz}$. The invariance conditions under $C_2(z)$ and $U_T \Theta \sigma_{yz}$ are given by $R_{\pi(z)}^\dagger M_{AA,(0,0)} R_{\pi(z)} = M_{AA,(0,0)}$ and $R_{\sigma(x)}^\dagger M_{AA,(0,0)}^* R_{\sigma(x)} = M_{AA,(0,0)}$, which result in the matrix form given in Eq. (24). This matrix also satisfies $U_T \Theta \sigma_{xz}$ invariance. By $S_4$ symmetry, the on-site potential for $B$-type dimers

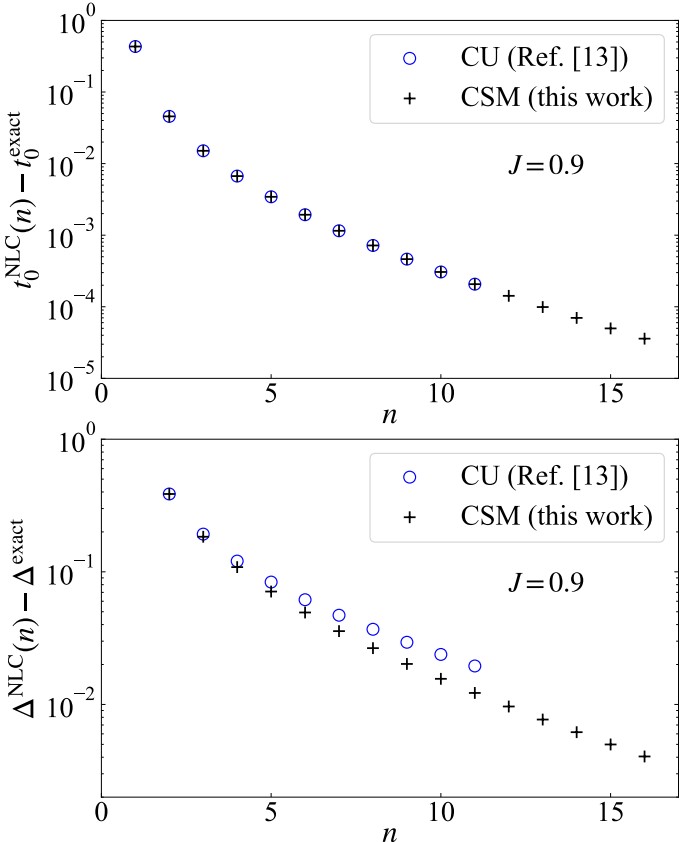

Figure 6: Comparison of NLC expansion results obtained using the CSM and continuous unitary (CU) transformations in block-diagonalization. Differences between the NLC expansion results and exact values are plotted. (Upper) On-site potential $t_0^{\mathrm{NLC}}(n)$. (Lower) Energy gap $\Delta^{\mathrm{NLC}}(n)$. Data for the CU transformation are adapted from Ref. [13].

433 becomes

$$M_{BB,(0,0)} = \begin{bmatrix} R_{02} & I_{01} & 0 \\ -I_{01} & R_{01} & 0 \\ 0 & 0 & R_{03} \end{bmatrix}. \tag{C.3}$$

434 All $R_i$ and $I_i$ in this appendix denote real and imaginary numbers, respectively.

435 For the nearest-neighbor hopping term $t^{\dagger}_{r+(1,0)} M_{BA,(1,0)} t_r$ ($r \in \Lambda_A$), $U_T \Theta \sigma_{xz}$ invariance

436 imposes Eq. (25). From the $U_T \Theta \sigma_{yz}$ and $C_2(z)$ symmetry, we obtain

$$M_{BA,(-1,0)} = \begin{bmatrix} I_{11} & R_{11} & -I_{12} \\ R_{12} & I_{13} & -R_{13} \\ -I_{14} & -R_{14} & I_{15} \end{bmatrix}. \tag{C.4}$$

437 $S_4$ symmetry further leads to

$$M_{AB,(0,1)} = \begin{bmatrix} -I_{13} & R_{12} & -R_{13} \\ R_{11} & -I_{11} & I_{12} \\ -R_{14} & I_{14} & -I_{15} \end{bmatrix}, \quad M_{AB,(0,-1)} = \begin{bmatrix} -I_{13} & R_{12} & R_{13} \\ R_{11} & -I_{11} & -I_{12} \\ R_{14} & -I_{14} & -I_{15} \end{bmatrix}. \tag{C.5}$$

438 For second-neighbor hopping, $M_{AA,(1,1)}$ and its Hermitian conjugate satisfy $C_2(z)$ symmetry,

439 which results in

$$M_{AA,(1,1)} = \begin{bmatrix} R_{21} & C_{21} & C_{22} \\ C^*_{21} & R_{22} & C_{23} \\ -C^*_{22} & -C^*_{23} & R_{23} \end{bmatrix}, \tag{C.6}$$

440 where $C_i$ denote complex numbers. From the $U_T \Theta \sigma_{yz}$ invariance, we have

$$M_{AA,(-1,1)} = \begin{bmatrix} R_{21} & -C^*_{21} & -C^*_{22} \\ -C_{21} & R_{22} & C^*_{23} \\ C_{22} & -C_{23} & R_{23} \end{bmatrix}. \tag{C.7}$$

441 $S_4$ symmetry further provides

$$M_{BB,(-1,1)} = \begin{bmatrix} R_{22} & -C^*_{21} & C_{23} \\ -C_{21} & R_{21} & -C_{22} \\ -C^*_{23} & C^*_{22} & R_{23} \end{bmatrix}, \quad M_{BB,(1,1)} = \begin{bmatrix} R_{22} & C_{21} & -C^*_{23} \\ C^*_{21} & R_{21} & -C^*_{22} \\ C_{23} & C_{22} & R_{23} \end{bmatrix}. \tag{C.8}$$

442 For third-neighbor hopping, $M_{AA,(2,0)}$ is invariant under both $U_T \Theta \sigma_{xz}$ and $C_2(z)$, and $S_4$

443 symmetry relates this to $M_{BB,(0,2)}$, which results in

$$M_{AA,(2,0)} = \begin{bmatrix} R_{31} & I_{31} & R_{32} \\ -I_{31} & R_{33} & I_{32} \\ -R_{32} & I_{32} & R_{34} \end{bmatrix}. \quad M_{BB,(0,2)} = \begin{bmatrix} R_{33} & I_{31} & I_{32} \\ -I_{31} & R_{31} & -R_{32} \\ I_{32} & R_{32} & R_{34} \end{bmatrix}. \tag{C.9}$$

444 Another third-neighbor hopping matrix $M_{AA,(0,2)}$ and its symmetry operation are given by

$$M_{AA,(0,2)} = \begin{bmatrix} R'_{31} & I'_{31} & I'_{32} \\ -I'_{31} & R'_{32} & R'_{33} \\ I'_{32} & -R'_{33} & R'_{34} \end{bmatrix}, \quad M_{BB,(2,0)} = \begin{bmatrix} R'_{32} & I'_{31} & -R'_{33} \\ -I'_{31} & R'_{31} & I'_{32} \\ R'_{33} & I'_{32} & R'_{34} \end{bmatrix}, \tag{C.10}$$

445 where $R'_i$ and $I'_i$ denote real and imaginary numbers, respectively.

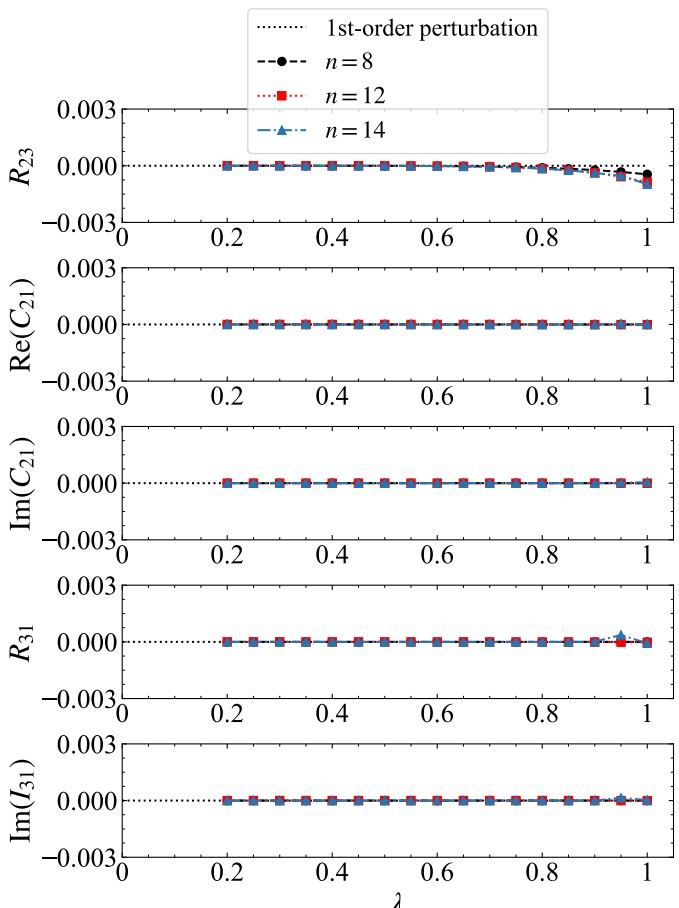

Figure 7: $\lambda$-dependence of selected matrix elements for second-neighbor hopping [$R_{23}$, Re($C_{21}$), and Im($C_{21}$)] and third-neighbor hopping ($R_{31}$ and $I_{31}$) at $J'/J = 0.6$ and $h/J = 0.015$. Dashed lines indicate the results of the first-order perturbative expansion. The parameter $n$ represents the number of spins in the largest clusters used in the NLC expansion.

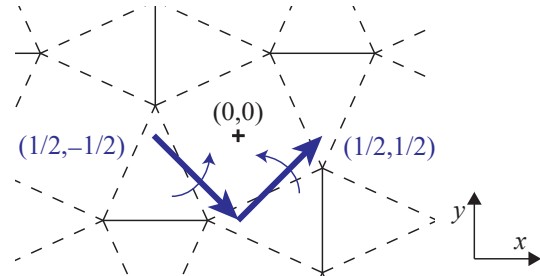

Figure 8: Translation vectors associated with $2_1$ Screw rotations.

## D    Analysis of $\lambda$-dependence in extended hopping matrix elements

Figure 7 presents the selected matrix elements of the hopping terms between second-neighbor and third-neighbor dimers as a function of the parameter $\lambda$. These results were obtained using the NLC expansion up to the $n$-spin clusters. While most matrix elements remain negligibly small, the diagonal components of the second-neighbor hopping term show a slight enhancement near $\lambda = 1$.

## E    Kramers degeneracy in the triplon bands of $SrCu_2(BO_3)_2$

As discussed in Appendix C, $SrCu_2(BO_3)_2$ in a magnetic field along the $z$-axis preserves symmetry under the operations $S_4$, $U_T\Theta\sigma_{xz}$, and $U_T\Theta\sigma_{yz}$. Consequently, the system is invariant under the combined operations $U_T\Theta\sigma_{xz}S_4$ and $U_T\Theta\sigma_{yz}S_4$. These operations act on both spatial coordinates and spin components, as follows:

$$U_T\Theta\sigma_{xz}S_4 : \begin{aligned} &(x,y,z) \rightarrow (-y,-x-1,-z),\\ &(S^x,S^y,S^z) \rightarrow (-S^y,-S^x,S^z), \end{aligned} \tag{E.1}$$

$$U_T\Theta\sigma_{yz}S_4 : \begin{aligned} &(x,y,z) \rightarrow (y+1,x,-z),\\ &(S^x,S^y,S^z) \rightarrow (S^y,S^x,S^z). \end{aligned} \tag{E.2}$$

The spatial part of each transformation is equivalent to that of a $2_1$ screw rotation, which consists of a translation along $(1/2,-1/2)$ or $(1/2,1/2)$ followed by a $\pi$-rotation about the corresponding axis (see Fig. 8). However, the associated spin transformations are different from those in the $2_1$ screw rotations.

At certain momenta, the combined operations become anti-unitary. When applied twice, they generate lattice translations in the $(1,-1)$ and $(1,1)$ directions:

$$(U_T\Theta\sigma_{xz}S_4)^2 = T_{(1,-1)}, \tag{E.3}$$

$$(U_T\Theta\sigma_{yz}S_4)^2 = T_{(1,1)}, \tag{E.4}$$

where $T_r$ denotes a translation by vector $r$. For Bloch eigenstates with momentum $k$, the translation operator yields a phase factor $\exp(ir \cdot k)$. Along the boundary of the Brillouin zone defined by the two-sublattice structure, where $k_x + k_y = (2n+1)\pi$ or $k_x - k_y = (2n+1)\pi$ for any integer $n$, this phase factor equals $-1$, indicating the anti-unitary character.

Anti-unitary symmetries are known to protect twofold degeneracies in the energy spectrum. Therefore, Kramers degeneracy enforces doubly degeneracy of triplon bands at these boundaries in momentum space. This result explains why, in $SrCu_2(BO_3)_2$, triplon bands are guaranteed to touch at the Brillouin zone boundary, resulting in symmetry-protected band crossings.

These symmetry considerations imply that, in $SrCu_2(BO_3)_2$, which has a quasi-two-dimensional structure, the excitation spectrum in three-dimensional momentum space exhibits symmetry-protected nodal planes. Similar Kramers degeneracies and nodal planes in spin excitations have been reported in a spin model for Volborthite [47] and in the Kitaev–Heisenberg model [48].

# F  Momentum-space representation of the effective triplon Hamiltonian

The effective Hamiltonian in momentum space is expressed as

$$\mathcal{H}_{\text{eff}} = \sum_{\alpha,\beta=A,B} \sum_{\boldsymbol{k}} t_{\boldsymbol{k},\alpha}^{\dagger} M_{\alpha\beta}(\boldsymbol{k}) t_{\boldsymbol{k},\beta}, \tag{F.1}$$

where $t_{\boldsymbol{k},\alpha}^{\dagger} = N^{-1/2} \sum_{\boldsymbol{r}\in\Lambda_\alpha} t_{\boldsymbol{r}}^{\dagger} e^{i\boldsymbol{k}\cdot\boldsymbol{r}}$ defines the Fourier-transformed triplon creation operators for sublattice $\alpha = A, B$ over $N$ unit cells.

In the following, we present only the on-site and nearest-neighbor hopping terms for clarity. However, in the numerical calculations, we also included second- and third-neighbor hopping processes to ensure quantitative accuracy.

The diagonal blocks $M_{\alpha\alpha}(\boldsymbol{k})$ are given by

$$M_{AA}(\boldsymbol{k}) = \begin{bmatrix} R_{01} & I_{01} & 0 \\ -I_{01} & R_{02} & 0 \\ 0 & 0 & R_{03} \end{bmatrix}, \quad M_{BB}(\boldsymbol{k}) = \begin{bmatrix} R_{02} & I_{01} & 0 \\ -I_{01} & R_{01} & 0 \\ 0 & 0 & R_{03} \end{bmatrix}. \tag{F.2}$$

The off-diagonal block $M_{AB}(\boldsymbol{k})$ is expressed as

$$2 \begin{bmatrix} -(I_{11}\cos k_x + I_{13}\cos k_y) & R_{12}(\cos k_x + \cos k_y) & i(-I_{14}\sin k_x + R_{13}\sin k_y) \\ R_{11}(\cos k_x + \cos k_y) & -(I_{11}\cos k_y + I_{13}\cos k_x) & i(-I_{12}\sin k_y + R_{14}\sin k_x) \\ i(-I_{12}\sin k_x + R_{14}\sin k_y) & i(-I_{14}\sin k_y + R_{13}\sin k_x) & -I_{15}(\cos k_x + \cos k_y) \end{bmatrix}, \tag{F.3}$$

with $M_{BA}(\boldsymbol{k}) = M_{AB}^{\dagger}(\boldsymbol{k})$.

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
