# Peer review of "Numerical Block-Diagonalization and Linked-Cluster Expansion for Deriving Effective Hamiltonians: Applications to Spin Excitations"

_SciPost Physics_

## Round 1 · Referee Report · Anonymous (Referee 1) · 2025-7-13

Strengths

1) Well written 2) Strength of the article is the treatment of avoided level crossings in the NLCE scheme 3) NLCE application to specific quantum spin models in one and two dimensions

Weaknesses

1) Novelty of introduced transformation 2) Explanation of clusters

Report

The article entitled "Numerical Block-Diagonalization and Linked-Cluster Expansion for Deriving Effective Hamiltonians: Applications to Spin Excitations" by Tsutomu Momoi and Owen Benton describes a variant of numerical linked-cluster expansions (NLCEs) for constructing effective low-energy Hamiltonians. The associated transformation is based on the Cederbaum-Schirmer-Meyer transformation allowing a block-diagonalization of the Hamiltonian. The interesting aspect of this transformation is that it is minimal with respect to changes in the basis states of the targeted low-energy subspace. The authors apply their transformation to the one-dimensional transverse-field Ising chain, which is exactly solvable and other NLCE approaches have been applied before, and the highly frustrated two-dimensional Shastry-Sutherland model in the presence of DM interactions. In both cases the dispersion of the one-particle excitations is calculated. In my opinion the article is nicely written and I find the research topic and the results very interesting. However, before I can recommend publication in SciPost Physics, I have the following list of comments and questions.

Specifically, let me address the following major points:

  • The authors should clarify the relation of their approach and the one in Ref. [17]. The latter introduces an NLCE based on projective cluster-additive transformation for quantum lattice models. Here already the minimal transformation is used to derive the effective Hamiltonian along the same lines as in the current work. One should further note that this minimal transformation is known in the literature under different names (Cederbaum, Takahashi, Schrieffer-Wolff, des Cloiseaux, ...) depending on the community. Further, Ref. [17] generalizes this transformation to arbitrary number of quasi-particle excitations (and not only one-particle excitations) and to models where there are processes coupling the ground-state and the one-particle sector, which is relevant to keep cluster additivity in the NLCE. In my opinion it is hard to argue that the current work represents the first which introduces this transformation to derive effective block-diagonal Hamiltonians for excitations of low-dimensional quantum spin models. I think the strength of the article is rather the treatment of avoided level crossings and the explicit application to specific quantum spin models in one and two dimensions (In [17] one- and two-particle excitations of the ordered phase of the 2D XXZ model on the square lattice have been calculated). The article should be revised accordingly.

  • In Section 2.3/2.4 it should be stressed that the NLCE only works in the described way if the single-particle sector is not directly coupled to the ground-state sector (and the same is true for many-particle excitations). This condition is indeed fulfilled for both applications described in the article. If this is not the case, one has to use the projections introduced in Ref. [17].

  • For the application of the method to the two-dimensional Shastry-Sutherland-model the authors describe the used clusters in Sec. 4.2.1 in words, which is not completely easy to capture. Can the author visualize the used clusters? In addition, how do the results depend on the chosen hierarchy of clusters, e.g., how does it compare to a full graph expansion truncated in different perturbative orders or rectangular graphs?

Minor points:

  • The authors use NLC expansion for the abbreviation of numerical linked cluster expansion. More common is NLCE in the literature.
  • Line 197 and below: The critical exponent is called "z" not large "Z". At the same time one maybe can change the letter for the coordination number.
  • Line 326: Formula in line is too long.
  • Eq. (27): Chern number is usually called large "C_n".

Recommendation

Ask for major revision

---

## Round 1 · Referee Report · Anonymous (Referee 2) · 2025-7-14

Strengths

1) Clear 2) Well-written 3) Effective Hamiltonian even in the case of avoided level crossing

Weaknesses

1) Could be more detailed on how this method is different from other projective block-diagonalization techniques.

2) Both examples are gapped with a unique ground state.

Report

Rooted in block-diagonalization and numerical linked-cluster expansion, the Authors present a numerical technique for deriving low-energy effective Hamiltonians in a non-perturbative manner.

A central step of this method is the modification of the minimal norm ||1-T || criterion of the block-diagonalization method introduced by Cederbaum, Schirmer, and Meyer to require only the states in the relevant subspace to change minimally. The variational constraint, || 1- T_(11) || =min, provides a unique projective transformation and guides the selection of the relevant eigenstates, even in the case of avoided level crossings, where more conventional perturbative techniques are deficient.

The Authors test the validity of their method on the transverse-field Ising model and the Shastry-Sutherland model extended with Dzyaloshinskii-Moriya interaction.

I find the introduced technique interesting, which will be useful to the broader community with potential follow-up works. The paper is extremely well-written and well-structured, with the specific examples providing additional insights and greatly contributing to the understanding of the numerical block-diagonalization combined with NLCE.
Therefore, I recommend publication in SciPost Physics and suggest that the authors consider the comments below.

Requested changes

1) Both examples are gapped, and the partitioning of the Hilbert space involves (1) the unique ground state, (2) the single-particle states corresponding to the noninteracting excitations and comprising the effective Hamiltonian, and (3) the multi-particle states, which are the neglected high-energy sector. Some comments on the applicability of this method for cases when the ground state is not unique or the excitations are gapless would be beneficial. Linked-cluster expansion allows for the computation of multi-particle excitations (e.g., two-particle continuum and bound states). How does the proposed variational constraint look for an effective Hamiltonian comprising multi-particle channels? In the case of SrCu2(BO3)2, for example, the bound states of triplets have been shown to be particularly important [Ref. 32].

2) In the example of SrCu2(BO3)2, an illustration of the 1st, 2nd, and 3rd neighbor paths and the corresponding smallest connected clusters would be great. Also, it’s hard to imagine how the larger clusters (the cluster with seven bonds in particular) are constructed without a schematic illustration.

3) The symmetry analysis of the on-site and hopping matrices for SrCu2(BO3)2 seems to be a bit disconnected from the proposed numerical technique. It’s unclear whether it is relevant for carrying out the numerical block-diagonalization combined with NLCE. The same goes for the triplet degeneracy at the zone boundary, which is a consequence of nonsymmorphic symmetries and should be independent of any (non)perturbative approaches that yield an effective Hamiltonian.

4) The Authors refer to the emergence of topological bands in SrCu2(BO3)2 with Chern numbers (2,-1,-1), (0,1,-1) and (1,1,-2), but it isn’t discussed for which parameters those develop (probably for some field values). Furthermore, the main text mentions the computation of thermal Hall conductivity and its comparison to the perturbative treatment, but there are no corresponding results, e.g., in the form of a plot, shown in the manuscript.

Recommendation

Publish (easily meets expectations and criteria for this Journal; among top 50%)

---

## Round 1 · Referee Report · Anonymous (Referee 3) · 2025-7-17

Strengths

  1. The paper proposes a new criterion to find an optimized low-energy effective Hamiltonian which is suitable to practical numerical simulations.
  2. The paper introduces a practical method to find an optimal low-energy Hilbert space on which the effective Hamiltonian is constructed.

Weaknesses

  1. The discussions on the two applications of the method, especially those on how to adapt their criterion and the state-selection principle to specific systems, are not detailed enough, considering that the original idea introduced in Ref.[18] is relatively unknown.

Report

The paper introduces a method for finding a low-energy effective Hamiltonian based on the variational criterion originally proposed in Ref.[18], and apply it to two specific examples to demonstrate how it works in actual many-body systems. The key idea is that instead of using the variational criterion on the entire Hilbert space (as has been done in Ref. [18]), we do it only within a particular sector. This is particularly suitable for correlated many-body systems, in which the full set of eigenstates/energies is usually not available. Therefore, I believe that the results presented in this paper will be potentially useful, providing a new powerful tool to calculate low-energy spectra and various dynamical quantities. Nevertheless, considering that the underlying idea and method given in Ref. [18] are not well-known (I spent some time to fully understand the results in Ref. [18]), I found that the discussions in sections 3 and 4 are not sufficiently detailed even to the experts in the field.
For the above reasons, I suggest that the authors consider the points listed below and revise the manuscript.

Requested changes

  1. I recommend the authors adding more discussions/explanations how their criterion [eq. (8)] is applied to select the optimal low-energy Hilbert space by the numerical data for a given problem (for instance, choosing the optimal block size is already a non-trivial problem in complex systems like the Shastry-Sutherland model). My impression is that the 1D transverse-field Ising model considered in section 3 is an ideal system to demonstrate the machinery, whereas the discussion presented there is rather sketchy.

  2. In the authors' blocking scheme, separating the Hamiltonian into two pieces H0 and H1 is crucial. Nevertheless, they are not explicitly defined in section 3.

  3. In section 2.4, they discuss how the hopping amplitudes decay with the distance by assuming a simple model form [eq. (15)] of the dispersion relation. The results [eqs. (16) and (17)] are reasonable from the standard lore of the Ornstein-Zernike theory. However, the spectrum assumed in eq. (15) seems rather specific. Are these results just to illustrate how the hopping amplitudes behave with the distance or to claim anything general?

  4. At the beginning of section 4.1.2, the authors say "Although DM interactions break the triplon number conservation, ..", which I do not think is very precise. Even without the DM interactions, the triplon number is never conserved unless J' is zero. Probably, they refer to the DM interaction in H0, as the conserved quantities of H0 are crucial to guess the correct block structure. Then, I do not understand how they correctly identify the block structure without having well-defined conserved quantities. They should elaborate more on this.

Recommendation

Ask for minor revision

---

## Editorial Decision

awaiting_resubmission